# Local Injection of Stem Cells Can Be a Potential Strategy to Improve Bladder Dysfunction after Outlet Obstruction in Rats

**DOI:** 10.3390/ijms25158310

**Published:** 2024-07-30

**Authors:** Ching-Chung Liang, Steven W. Shaw, Tse-Ching Chen, Yi-Hao Lin, Yung-Hsin Huang, Tsong-Hai Lee

**Affiliations:** 1Female Urology Section, Department of Obstetrics and Gynecology, Chang Gung Memorial Hospital, Linkou Medical Center, Taoyuan 333, Taiwan; ccjoliang@cgmh.org.tw (C.-C.L.); linyihaou@yahoo.com.tw (Y.-H.L.); uoncin@cgmh.org.tw (Y.-H.H.); 2College of Medicine, Chang Gung University, Taoyuan 333, Taiwan; dr.shaw@me.com (S.W.S.); ctc323@cgmh.org.tw (T.-C.C.); 3Division of Obstetrics, Department of Obstetrics and Gynecology, Taipei Chang Gung Memorial Hospital, Taipei 105, Taiwan; 4Prenatal Cell and Gene Therapy Group, Institute for Women’s Health, University College London, London WC1E 6BT, UK; 5Department of Anatomical Pathology, Chang Gung Memorial Hospital, Linkou Medical Center, Taoyuan 333, Taiwan; 6Stroke Center and Department of Neurology, Chang Gung Memorial Hospital, Linkou Medical Center, Taoyuan 333, Taiwan

**Keywords:** stem cell, bladder outlet obstruction, bladder overactivity, detrusor, rat

## Abstract

This study investigates whether hAFSCs can improve bladder function in partial bladder outlet obstruction (pBOO) rats by targeting specific cellular pathways. Thirty-six female rats were divided into sham and pBOO groups with and without hAFSCs single injection into the bladder wall. Cystometry, inflammation/hypoxia, collagen/fibrosis/gap junction proteins, and smooth muscle myosin/muscarinic receptors were examined at 2 and 6 weeks after pBOO or sham operation. In pBOO bladders, significant increases in peak voiding pressure and residual volume stimulated a significant upregulation of inflammatory and hypoxic factors, TGF-β1 and Smad2/3. Collagen deposition proteins, collagen 1 and 3, were significantly increased, but bladder fibrosis markers, caveolin 1 and 3, were significantly decreased. Gap junction intercellular communication protein, connexin 43, was significantly increased, but the number of caveolae was significantly decreased. Markers for the smooth muscle phenotype, myosin heavy chain 11 and guanylate-dependent protein kinase, as well as M2 muscarinic receptors, were significantly increased in cultured detrusor cells. However, hAFSCs treatment could significantly ameliorate bladder dysfunction by inactivating the TGFβ-Smad signaling pathway, reducing collagen deposition, disrupting gap junctional intercellular communication, and modifying the expressions of smooth muscle myosin and caveolae/caveolin proteins. The results support the potential value of hAFSCs-based treatment of bladder dysfunction in BOO patients.

## 1. Introduction

Bladder outlet obstruction (BOO) commonly occurs in prostate hyperplasia [1], neurogenic bladder dysfunction, and postoperative conditions related to urinary incontinence and uterovaginal prolapse [2,3]. BOO can lead to a spectrum of urinary tract dysfunctions, encompassing overactive bladder syndrome, renal impairment, and urinary retention. Animal studies have revealed that high intravesical pressure within BOO-affected bladders can cause bladder hypoxia, inciting subsequent inflammatory cascades and profibrotic responses [4,5,6]. Ischemia and collagen deposition are notable features observed in the BOO-affected bladder, leading to smooth muscle cell hypertrophy, bladder wall fibrosis [6,7], and loss of contractility [6,8].

Urinary tract dysfunction resulting from BOO currently lacks effective treatment options. In vivo studies have demonstrated that extracted compounds from Salvia miltiorrhiza can improve bladder fibrosis in BOO animals via inhibiting activation of the TGFβ-Smad signaling pathway [9]. Additionally, soluble guanylate cyclase stimulators exhibit the ability to elevate cyclic guanosine monophosphate levels to improve detrusor overactivity [10]. Stem cell treatment of BOO animals has shown remarkable outcomes, improving urodynamic results and significantly reducing inflammatory mediators, thereby preventing or reversing bladder fibrosis [7,11,12]. While several studies have demonstrated that the ultrastructure of detrusor cells, such as gap junction intercellular communication and caveolae/caveolin [13,14], as well as myosin heavy chain isoforms in detrusor smooth muscle [15], is all related to BOO, these post-treatment ultrastructural changes remain unexplored. The present study aims to explore whether treatment with human amniotic fluid stem cells (hAFSCs) can improve bladder dysfunction via inducing ultrastructural changes in detrusor cells in BOO-affected rats.

## 2. Results

### 2.1. Post-pBOO Dysfunction as Determined by Cystometry Improved by hAFSCs Treatment

Compared with the sham-operated rats, partial BOO (pBOO) rats induced a significant increase in bladder weight and residual volume at 2 and/or 6 weeks post-pBOO induction. Additionally, there was a significant increment in the peak voiding pressure and bladder capacity but a significant decrement in the peak voided volume observed at 6 weeks following pBOO induction. However, treatment with hAFSCs resulted in significant improvements in bladder weight and these cystometrical parameters indicative of bladder dysfunction at 2 and/or 6 weeks post-treatment (Figure 1).

### 2.2. Expression of TGFβ-Smad Pathway Factors, Bladder Collagen Deposition, and Fibrosis Markers Improved by hAFSCs Treatment

Compared with sham-operated rats, expression levels of connexin 43, collagens 1 and 3, Smad2/3, and transforming growth factor (TGF)-β1 exhibited significant increases at 2 and 6 weeks following pBOO induction. However, treatment with hAFSCs led to significant improvements in the expression of these markers (Figure 2). Conversely, caveolins 1 and 3 exhibited significantly decreased expression at 2 and/or 6 weeks post-pBOO induction, with only caveolin 1 showing a significant increase following hAFSCs treatment.

### 2.3. Electron Microscopy of Caveolae in Detrusor Smooth Muscle Cells (DSMCs) Recovered after hAFSC Treatment

Electron micrographs showed a significant reduction in the number of caveolae in the bladder affected by pBOO at 2 and 6 weeks post-induction in contrast to the sham-operated rats. However, following treatment with hAFSCs, there was a notable increase in the number of caveolae (Figure 3).

### 2.4. Survival of hAFSCs in the pBOO Bladder

The tracking of implanted ethynyl deoxyuridine (EdU)-labeled hAFSCs within the bladder showed that the percentage of EdU-positive hAFSCs at days 3, 7, 14, and 28 was significantly lower as compared to the initial measurement on day 1 (Figure 4).

### 2.5. Immunofluorescent Expression of MYH11 and Receptors Mediating Bladder Contraction on DSMCs

Expressions of myosin heavy chain 11 (MYH11), guanylate-dependent protein kinase (PKG), and M2 muscarinic receptors (M2) in DSMCs significantly increased at 6 weeks post-pBOO induction but significantly improved after hAFSCs treatment (Figure 5). Conversely, α-SMA, M3 muscarinic receptors, tachykinin NK2, and purinergic P2X1 receptor transcripts showed no significant changes.

## 3. Discussion

The present study demonstrates that in pBOO bladders, significant increases in peak voiding pressure and residual volume stimulate significant upregulation of the inflammatory and hypoxic factors, TGF-β1 and Smad2/3. Subsequently, the collagen deposition proteins, collagen 1 and 3, were significantly increased, but the bladder fibrosis markers, caveolin 1 and 3, were significantly decreased. In addition, the gap junction intercellular communication protein, connexin 43, was significantly increased, but the number of caveolae was significantly decreased. Furthermore, markers for the smooth muscle phenotype, myosin heavy chain 11 and guanylate-dependent protein kinase, as well as M2 muscarinic receptors, were significantly increased in cultured detrusor cells. However, hAFSCs treatment could significantly ameliorate bladder dysfunction through the inactivation of the TGFβ-Smad signaling pathway, reduction in collagen deposition, disruption of gap junctional intercellular communication, and modifications in the expressions of smooth muscle myosin and caveolae/caveolin proteins.

In the present study, the residual volume of pBOO rats was observed to be higher than that of sham-operated rats at 2 weeks post-pBOO induction with a significant increase noted at 6 weeks. However, injection of hAFSCs into the bladder wall resulted in improvements in bladder weight and bladder functions, including peak voiding pressure, peak voided volume, residual volume, and bladder capacity in pBOO rats. Several preclinical studies have demonstrated the potential of mesenchymal stem cells in ameliorating bladder dysfunction in pBOO rats [11,12,16,17]. For instance, the transplantation of human bone marrow mesenchymal stem cells (BMSCs) into pBOO bladders has been shown to inhibit bladder fibrosis and restore peak voiding pressure and residual volume to normal levels [11]. Additionally, intravenous administration of rat BMSCs into pBOO rats has been found to normalize bladder capacity and significantly reduce levels of TGFβ levels and other inflammatory and profibrotic markers [12,16]. Notably, in pBOO bladders, the increased expression of collagen and TGFβ observed at 2 and 6 weeks post-pBOO induction was restored following the transplantation of human BMSCs [11,17].

Experimental studies have shown that bladder smooth muscle initially responds to hypoxia after pBOO, resulting in a hypoxic cascade including inflammation, profibrotic changes, and increased extracellular matrix expression [4,6]. The present study showed that the protein levels of inflammatory and hypoxia factors, namely TGF-β1 and Smad2/3, exhibit upregulation in pBOO rats at 2 weeks post-induction, followed by a gradual decline by 6 weeks. Prior research has underscored the role of augmented TGF-β1 in activating the TGFβ-Smad signaling pathway [18] and promoting bladder fibrosis secondary to pBOO [9]. It is reported that sodium transrhinone IIA sulfonate extracted from *Salvia miltiorrhiza* was found to improve bladder fibrosis in pBOO by inhibiting the activation of the TGFβ-Smad signaling pathway [9]. Collagen 1 and 3 emerge as pivotal interstitial collagens in the process of obstructive bladder fibrosis [6,9]. Increased collagen may reduce bladder wall contractility and compliance. Animal studies have indicated an elevation in total collagen and gene expression of collagens 1 and 3 post-BOO, followed by a decline upon relief of obstruction [19]. Notably, protein analysis in human studies revealed an increased level of total collagen, collagen 3/collagen 1 ratio, and collagen 3 in noncompliant bladders [20]. In the present study, protein expression levels of collagens 1 and 3 were increased at 2 and 6 weeks after pBOO induction but showed improvement after hAFSCs treatment.

Connexin 43, a gap junctional intercellular communication protein, is the major connexin expressed by bladder smooth muscle cells [21]. A previous study demonstrated that connexin 43 plays a significant role in the bladder wall’s response to increased voiding pressure during BOO [21]. Experimental studies have shown that connexin 43 mRNA expression is increased 6 weeks after pBOO induction [22]. In pBOO rat bladders, there was an observed upregulation of connexin 43 and a concurrent decrease in its cell membrane expression within the chronically stretched detrusor muscle [13]. The present study revealed that connexin 43 expression significantly increased at 2 and 6 weeks post-pBOO induction, with this upregulation notably enhanced following treatment with hAFSCs.

Caveolae, nonclathrin-coated plasma membrane invaginations, are associated with transcytosis, potocytosis, sorting of surface proteins, and the localization of signaling molecules [23]. Pathological disruption of caveolae or alterations in caveolar abundance could markedly impact bladder contractility, thereby precipitating bladder dysfunction [24]. In the present study, the electron micrograph analysis illustrated a decrease in the number of caveolae at 2 and 6 weeks following pBOO induction compared to control rats. However, this reduction was mitigated after hAFSCs treatment. Caveolin 1 serves as a principal component of caveolae, and changes in caveolar numbers typically correlate with changes in caveolin 1 levels [23,25]. Caveolin 1 was identified within the cytoplasmic membrane of urethral and bladder smooth muscle cells, while caveolin-3 was localized to the membrane of striated muscles within the intrinsic sphincter [26]. In an animal study, bladder smooth muscle of caveolin-1 knockout mice showed reduced caveolar number [27]. Our previous study showed that compared to non-acute urine retention rats, the expression of caveolin-1 and the number of caveolae in the bladder were significantly reduced in acute urine retention rats at 3 days postpartum [28]. Hypertrophied detrusor smooth muscle in the bladder under pBOO conditions exhibited a reduced number of caveolae, disordered distribution at the plasma membrane, and diminished protein expression of caveolins 1 and 3 [14]. The present study showed that the expression of caveolins 1 and 3 decreased at 2 and/or 6 weeks post-pBOO induction, but only caveolin 1 was significantly increased after hAFSCs treatment.

In this study, we found the amount of hAFSCs is decreased almost to zero at week 2 (Figure 4) after local injection to the bladder wall, but hAFSCs treatment can still improve the bladder dysfunction at post-pBOO week 6 (Figure 1). It is likely that the mechanism of hAFSCs for the functional improvement following pBOO is not only due to neurogenesis and angiogenesis caused by the differentiation of injected stem cells but also by multimodal actions. Recently, Teng et al. [29] have suggested a concept of functional multipotency of mesenchymal stem cells. The functional multipotency includes the capabilities to (1) shed secretomes and exosomes containing many growth factors, anti-inflammatory proteins, membrane receptors, and microRNAs; (2) perform chemotactic migration towards developmental destinies or inflammatory areas; (3) recruit endogenous stem cells; (4) form gap junction and cell fusion; and (5) initiate asymmetric cell division and neural lineage differentiation [29]. This multipotency enables stem cells to interact with the surrounding environment and maximize their survival.

Our results showed that expressions of MYH11 and PKG in DSMCs were enhanced at 6 weeks post-pBOO induction, but these levels improved following hAFSCs treatment. MYH11 generates a myosin heavy chain protein, showing specific expression within smooth muscle cells. The expression of smooth muscle myosin (SMM) and actin isoforms serves as a molecular marker of smooth muscle differentiation, including bladder muscle [30]. A previous study has shown that reduced smooth muscle contractile activity in pBOO rabbit bladders is associated with the changes in the protein and mRNA expression of SMM heavy chain isoforms SM1 and SM2 [31]. In addition to SMM expression, the expression of PKG is also regarded as another pivotal marker of the smooth muscle phenotype. A previous report has shown that a cell line derived from hypertrophic rabbit bladder smooth muscle can express PKG [30]. An in vivo study has shown that activation of the nitric oxide-cGMP pathway via PKG reduces phasic contractions in neonatal rat bladder strips [32].

The present study additionally observed an increase in the expression of M2 receptors in DSMCs at 6 weeks post-pBOO induction, which significantly decreased following hAFSCs treatment. A previous study has demonstrated that, in comparison to control rats, the protein expression of the M2 receptor in both urothelium and detrusor muscle exhibited a significant increase 3 weeks after pBOO induction [33]. Other principal receptors mediating bladder contraction include muscarinic receptors, tachykinin, and purinergic receptors [31]. However, under culture conditions, bladder cells may experience a reduction in receptors involved in contraction. This reduction includes significant decreases in the M3 muscarinic receptor, tachykinin NK2, and purinergic P2X1 receptor transcripts, as their contraction function is not required in cultured conditions [34].

The present study has several limitations. Firstly, it focused on functional and morphological changes in pBOO bladders only at 2 and 6 weeks post-hAFSCs treatment. Cystometric analysis may yield different results if performed longer after hAFSCs treatment. Furthermore, this study did not assess immunosuppression; however, our previous demonstration in animal models showed that injection of hAFSCs did not produce any immune rejection of transplanted tissue [7,35].

## 4. Materials and Methods

### 4.1. Experiment Design

All animal care and experimental protocols were conducted in accordance with the guidelines established by the Institutional Ethics Committee for the Care and Use of Experimental Animals (approval no. 2020121901) and the Institutional Review Board of our institute (approval no. 202002270A3). The quality of the published reports of animal experiments was assessed using the ARRIVE guidelines. Thirty-six female Sprague–Dawley rats (10–12 weeks old) were randomly assigned into sham-operated rats (control group), pBOO rats treated with a bladder wall injection of 0.3 mL phosphate-buffered saline (pBOO + PBS), or pBOO rats that received a bladder wall injection of 1 × 1,000,000 hAFSCs in 0.3 mL PBS (pBOO + hAFSCs). All rats underwent a bladder function test by conscious cystometry at 2 and 6 weeks post-pBOO induction (N = 6 per time point). After cystometry, the rats were euthanized, and the bladders were excised and weighed. The expressions of connexin 43 protein, collagens 1 and 3, TGF-β1, Smad2/3, and caveolins 1 and 3 in the bladder were assessed through Western blot analysis. The quantification of caveolae was performed by using electron microscopy. For the in vitro study, immunofluorescence was used to analyze the expressions of α-smooth muscle actin, MYH11, cyclic PKG, M2 and M3 muscarinic receptors, the purinergic P2X1 receptor, and the tachykinin NK2 receptor in cultured detrusor smooth muscle cells (DSMCs) in each group. The experimental workflow is outlined in Figure 6.

### 4.2. Animal Models of pBOO

Partial BOO was induced in rats via proximal urethral ligation as described in the previous study [36]. Under 2% isoflurane inhalation, the abdominal cavity of the rat was opened through a suprapubic midline incision to expose the urethravesical junction. A polyethylene 50 tube was inserted into the urethra to serve as a stent for urethral ligation, resulting in an approximate outer diameter of 1.0 mm for the catheterized urethra. Subsequently, a 5–0 silk suture was gently looped around the catheterized urethra, followed by prompt removal of the catheter.

The reasons why the present study used female rats are that, first, most authors are clinicians and experts in female urology and obstetrics. The use of female rats in animal studies can help to understand and solve similar urological problems in female patients with pBOO. Second, our previous studies used female rat models with diabetes and urine retention [28,35] to examine the protective effect of hAFSCs against bladder dysfunction. In order to keep study consistency, we intend to examine if hAFSCs treatment can also improve bladder dysfunction in pBOO rats similar to the female rats with diabetes and urine retention.

### 4.3. Isolation and Characterization of hAFSCs for Treatment

The hAFSCs were derived from freshly collected amniotic fluid obtained via routine amniocentesis from second-trimester healthy pregnant donors. Using flow cytometry, the specific surface antigens of hAFSCs were characterized, as outlined in our previous study [35]. The cells in culture were trypsinzed and stained with phycoerythrin (PE)-conjugated antibodies against CD44, CD73, CD90, CD105, CD117, and CD45 (BD PharMingen, San Diego, CA, USA). Thereafter, the cells were analyzed using the FACS Calibur flow cytometer (Becton Dickinson, Heidelberg, Germany). These cells showed positive expression of CD44, CD73, CD90, CD105, and CD117 but were negative for CD45. Passage 6–8 hAFSCs were collected and prepared to a final concentration of 1 × 1,000,000 cells per 0.3 mL in PBS. In the groups receiving hAFSCs treatment, 1 × 1,000,000 ethynyl deoxyuridine (EdU)-labeled hAFSCs were injected into 5 sites of the submucosal layer of the bladder wall (anterior, posterior, bilateral, and dome) using a 500 μL syringe with a 26-gauge needle according to our previous method [35]. During each local injection, the syringe was pushed backward to confirm the needle was not present inside the vessel and was inside the muscular tissue of the bladder.

### 4.4. Conscious Cystometry

Two days post-implantation of the suprapubic catheter, the animals were placed in a specialized metabolic cage (Med Associates Inc., St. Albans, VT, USA) for conscious cystometry at 2 and 6 weeks after sham, pBOO + PBS, and pBOO + hAFSCs treatments, following protocols established in our prior studies [35]. All cystometric parameters were collected for analysis over 5 consecutive micturition cycles. Cystometric analysis was performed using Cystometry Analysis Version 1.05 (Catamount Research and Development, St. Albans, VT, USA), as per established protocols.

### 4.5. Western Blot Analysis

Total protein was isolated from bladder tissue, and its concentration was determined using a BCA protein assay kit (Thermo Scientific, Waltham, MA, USA) as described in a previous study [9]. Proteins were isolated from the samples by SDS-PAGE electrophoresis and subsequently transferred onto a PVDF membrane. Following blocking with 5% non-fat milk, the membrane was incubated overnight at 4 °C with rabbit polyclonal antibodies against connexin 43, Smad2/3, and caveolins 1 and 3, as well as mouse monoclonal antibodies against collagens 1 and 3, TGF-β1, and β-actin (Appendix A). The secondary antibodies were goat anti-rabbit IgG and goat anti-mouse IgG (1:5000, Cell Signaling Technology, Inc., Danvers, MA, USA) diluted in TBST solution.

The signals were scanned and visualized by UVP Chemstudio (Analytic Jena, Jena, Germany). The Western strap density was quantified using ImageJ software (Version 1.47), with grayscale values used to represent the expression levels of the target protein. Relative protein levels were determined by comparing gray values of the target proteins with those of β-actin.

### 4.6. Culture of DSMCs

The tissue from the bladder body was cooled on ice, followed by precise dissection to remove the mucosa and serosa layers covering the detrusor muscle using scissors. In a shaking incubator at 37 °C, the smooth muscle layer was incubated in PBS (−) containing 0.2% trypsin for 30 min, as described in a previous study [37]. Minced with scissors, the smooth muscle layer was suspended in RPMI-1640 medium containing 0.1% collagenase. Following digestion, the cell suspension was incubated for 30 min at 37 °C. It was then centrifuged at 250× *g* to separate contaminating tissue, including intimal cells, fibroblasts, and tissue debris. Smooth muscle cells were collected from the supernatant. Subsequently, the pure smooth muscle cells were cultured in a dish containing RPMI-1640 supplemented with 10% fetal bovine serum within a humidified atmosphere of 95% and 5% air and CO_2_, respectively. The culture medium was changed every 3 days. On the seventh day, the rat bladder body smooth muscle cell culture was subjected to immunostaining using an anti-α-actin antibody.

### 4.7. Immunofluorescence

Immunofluorescence was employed to analyze the expressions of α-smooth muscle actin, MYH11, and cyclic PKG, as well as M2 and M3 muscarinic receptors, the purinergic P2X1 receptor, and the tachykinin NK2 receptor, in cultured DSMCs across all experimental groups (Appendix A). Passage 2–3 DSMCs at an appropriate density were collected and attached to 12-well plates. Cells were treated with 4% paraformaldehyde for 5 min, followed by 3 PBS washes. After incubating with a 1% bovine serum albumin blocking reagent and 0.1% Triton-X 100, cells underwent washing and were then exposed to primary antibodies for 20 h at 4 °C, followed by incubation with secondary antibodies Alexa-flor 488 and 594 (1:250, Invitrogen, Grand Island, NY, USA) for 1 h. Nuclear staining was conducted using 4′,6-diamidino-2-phenylindole (DAPI), and subsequent immunofluorescence analysis was performed using an Olympus BX-51 microscope equipped with Image-Pro Plus software (Version 6.0, Media Cybernetics, Silver Spring, MD, USA).

HAFSCs implanted in the bladder were tracked by calculating the percentage of EdU-positive hAFSCs based on nuclear counts in DAPI- and EdU-stained images. For all reactions, immunoreactivity ratio levels in pBOO rats with or without hAFSCs treatment were determined relative to that of sham-operated rats.

### 4.8. Electron Microscopy

Bladder tissue specimens were initially preserved by immersion in a solution containing 4% glutaraldehyde buffered with cacodylate (0.15 M, pH 7.2) for electron microscopy studies. Following this fixation step, the samples underwent further stabilization through immersion in a 1% osmium tetroxide solution (0.1 M, pH 7.2). Dehydration of the specimens was then carried out using a series of graded ethanol and propylene oxide solutions. Finally, the dehydrated tissues were embedded in Spurr’s resin (manufactured by EMS, Hatfield, PA, USA) to facilitate their examination under the electron microscope. Ultrathin sections, approximately 90 nm thick, were prepared using an RMC ultramicrotome MT-7000 (RMC, Tucson, AZ, USA) and affixed onto copper grids. Subsequently, these sections underwent staining with 10% uranyl acetate and lead citrate to enhance contrast. Ultrastructural analysis was conducted utilizing a HT7800 transmission electron microscope (TEM, Hitachi 7800, Tokyo, Japan) at ×54,000. The quantification of caveolae within the bladder smooth muscle was performed by counting their occurrences in four randomly selected areas measuring 2.5 cm^2^ each. The mean of caveolae observed within bladder smooth muscle cells was utilized for statistical analysis.

### 4.9. Statistical Analysis

Data analysis was conducted utilizing Prism 5 (GraphPad Software, Inc., San Diego, CA, USA), with continuous variables presented as mean ± standard deviation (SD). One-way analysis of variation was employed to compare continuous data across different groups, followed by post hoc comparisons using The Tukey–Kramer test. The effect of hAFSCs within each group was assessed through chi-square testing with Fisher’s exact test. Statistical significance was defined as probability values < 0.05.

## 5. Conclusions

The present study marks the first investigation to report ultrastructural changes within detrusor cells in a rat pBOO model subsequent to hAFSCs treatment. Our findings indicate the potential therapeutic efficacy of hAFSCs treatment in ameliorating bladder detrusor dysfunction in pBOO rats. This improvement appears to be associated with several underlying mechanisms, including the inactivation of the TGFβ-Smad signaling pathway, reduction in collagen deposition, disruption of gap junctional intercellular communication via connexin 43, and variations in expression patterns of smooth muscle myosin and caveolae density/caveolin.

## Figures and Tables

**Figure 1 ijms-25-08310-f001:**
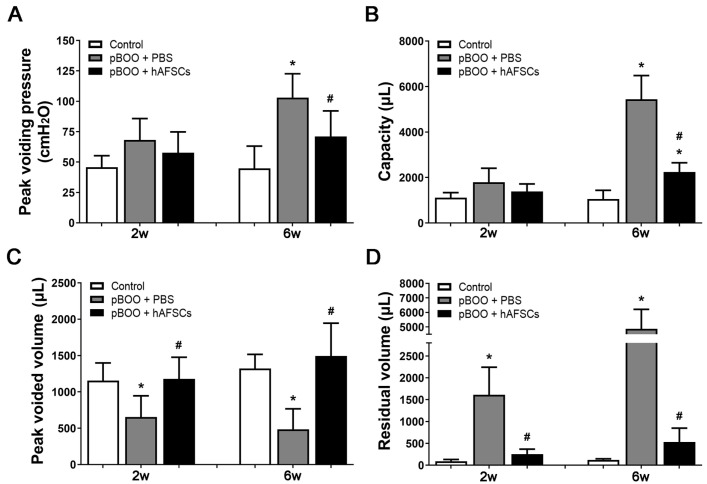
Cystometric results and bladder weight. Cystometric variables include peak voiding pressure (**A**), bladder capacity (**B**), peak voided volume (**C**), and residual volume (**D**). Compared with control rats, pBOO rats induced a significant increase in bladder weight and residual volume at 2 and/or 6 weeks following pBOO induction. Additionally, there was a significant increment in the peak voiding pressure and bladder capacity, but a significant decrement in the peak voided volume observed at 6 weeks following pBOO induction. However, bladder dysfunction and bladder weight in the pBOO rats can be improved after hAFSCs treatment at 2 and/or 6 weeks post-pBOO induction. *: *p* < 0.05 vs. control, #: *p* < 0.05 vs. pBOO + PBS. N = 6 at each time point. hAFSCs = human amniotic fluid stem cells; PBS = phosphate-buffered saline; pBOO = partial bladder outlet obstruction.

**Figure 2 ijms-25-08310-f002:**
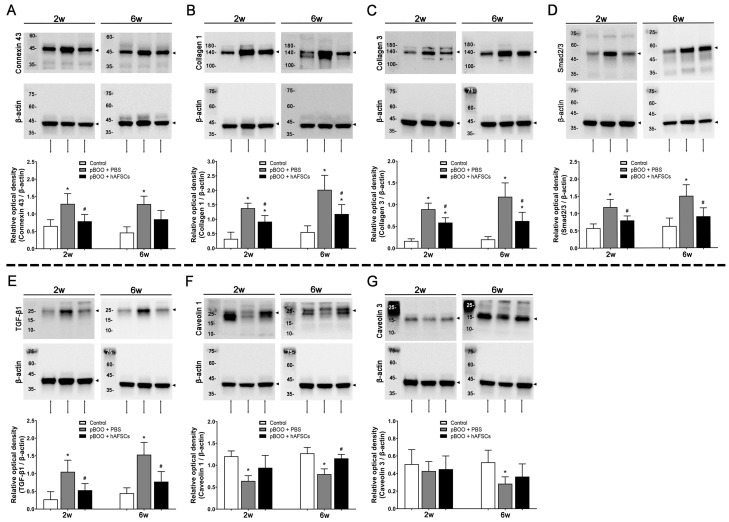
Western blot analysis of connexin 43 (**A**), collagen 1 (**B**), collagen 3 (**C**), Smad2/3 (**D**), TGF-β1 (**E**), caveolin 1 (**F**), and caveolin 3 (**G**) protein expression in bladder tissue. Protein expression of connexin 43, collagen 1, collagen 3, Smad2/3, and TGF-β1 is significantly increased at 2 and/or 6 weeks post-pBOO induction but improved after hAFSCs treatment. The protein expression of caveolin 1 and 3 is decreased at 2 and/or 6 weeks post-pBOO induction, but only caveolin 1 is improved significantly after hAFSCs treatment. *: *p* < 0.05 vs. control, #: *p* < 0.05 vs. pBOO + PBS. N = 6 at each time point. hAFSCs = human amniotic fluid stem cells; PBS = phosphate-buffered saline; pBOO = partial bladder outlet obstruction; TGF-β1 = transforming growth factor-β1.

**Figure 3 ijms-25-08310-f003:**
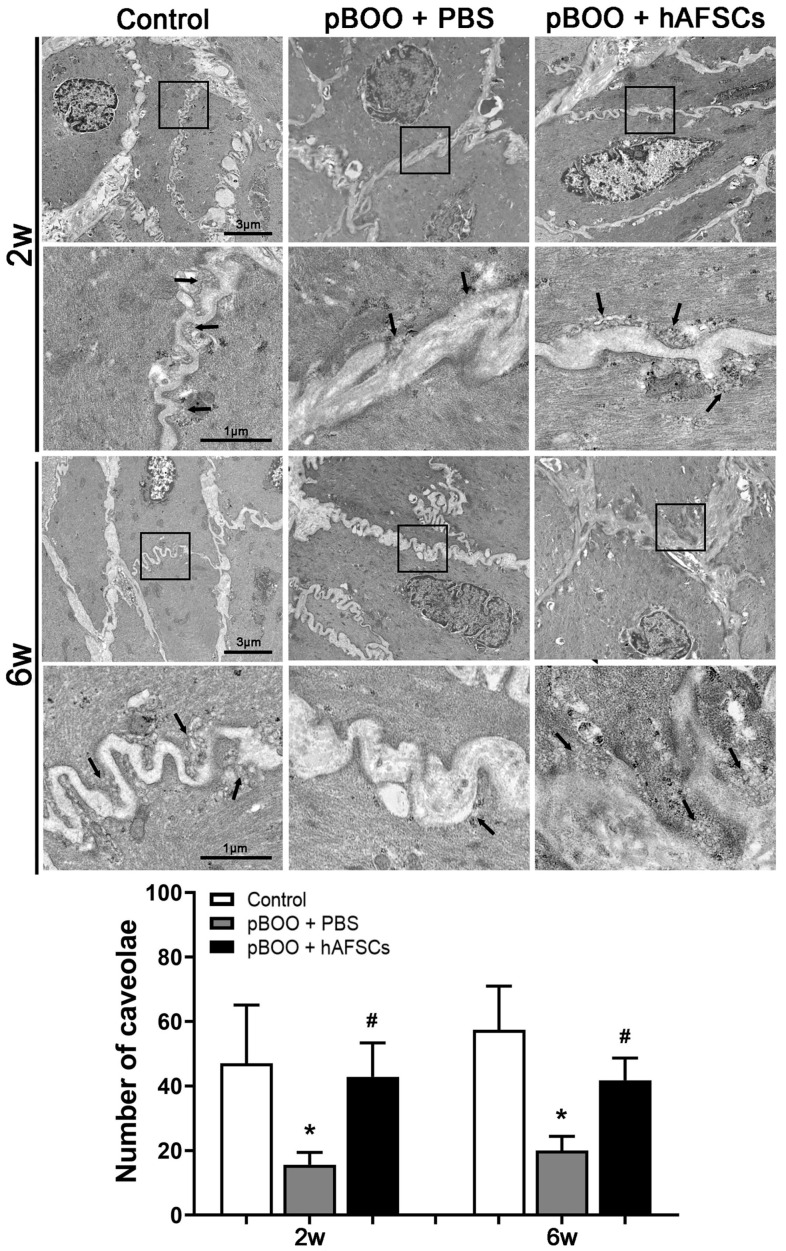
Electron micrographs of bladder smooth muscle cells in the control, pBOO + PBS, and pBOO + hAFSCs rats. Electron micrographs show that compared with the control rats, caveolae (black arrow) significantly decreased at 2 and 6 weeks post-pBOO induction but returned to the control level after hAFSCs treatment. Bar indicates 1 or 3 μm in electron micrographs. *: *p* < 0.05 vs. control, #: *p* < 0.05 vs. pBOO + PBS. N = 6 at each time point. hAFSCs = human amniotic fluid stem cells; PBS = phosphate-buffered saline; pBOO = partial bladder outlet obstruction. The black boxes indicate the area with high magnification (×54,000) placed below.

**Figure 4 ijms-25-08310-f004:**
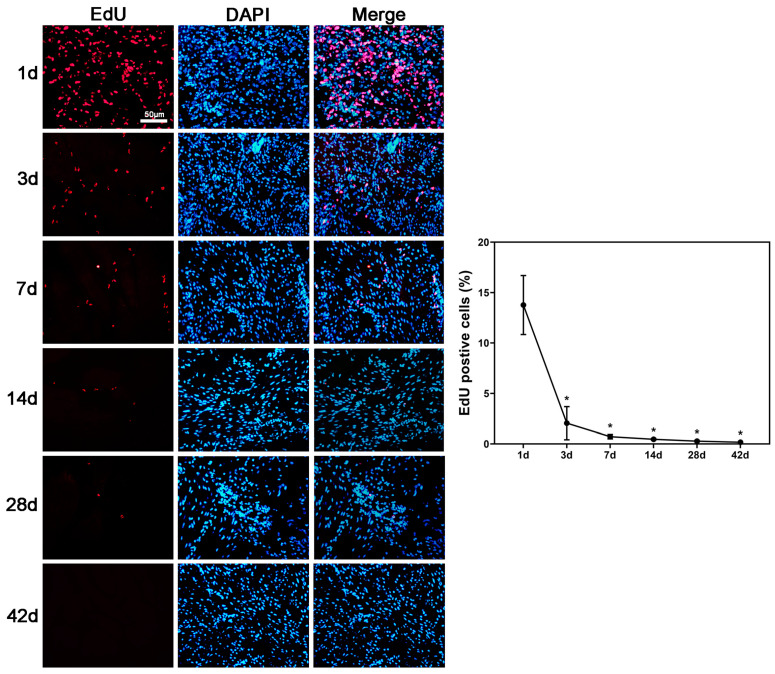
Tracking of implanted hAFSCs. Immunofluorescent imaging reveals the presence of implanted ethynyl deoxyuridine (EdU)-labeled hAFSCs in the bladder. The number of EdU-positive hAFSCs on days 3, 7, 14, and 28 is significantly lower than that on day 1. Bar in EdU = 50 μm. *: *p* < 0.05 vs. control. N = 6 at each time point. DAPI = 4′,6-diamidino-2-phenylindole; hAFSCs = human amniotic fluid stem cells.

**Figure 5 ijms-25-08310-f005:**
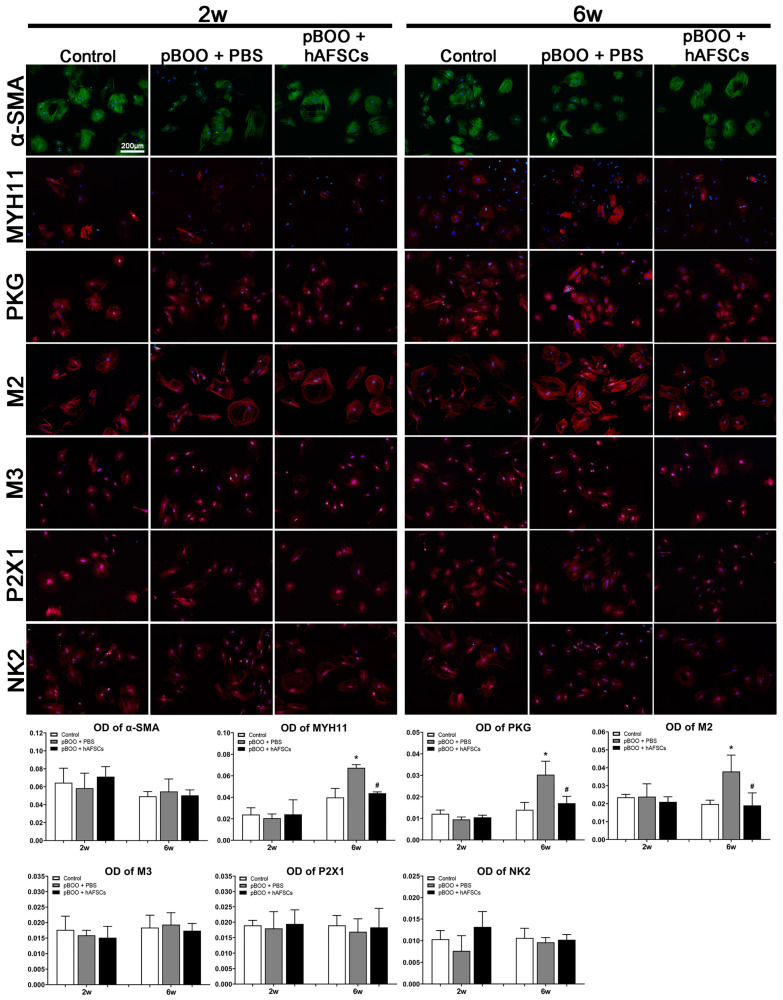
Temporal immunofluorescent expressions of α-SMA, MYH11, PKG, M2, M3, P2X1, and NK2 in cultured detrusor smooth muscle cells (DSMCs). Compared with the control rats, only the expressions of MYH11, PKG, and M2 in DSMCs increased significantly at 6 weeks post-pBOO induction but improved after hAFSCs treatment. *: *p* < 0.05 vs. control, #: *p* < 0.05 vs. pBOO + PBS. N = 6 at each time point. Bar indicates 200 μm. hAFSCs = human amniotic fluid stem cells; pBOO = partial bladder outlet obstruction.

**Figure 6 ijms-25-08310-f006:**
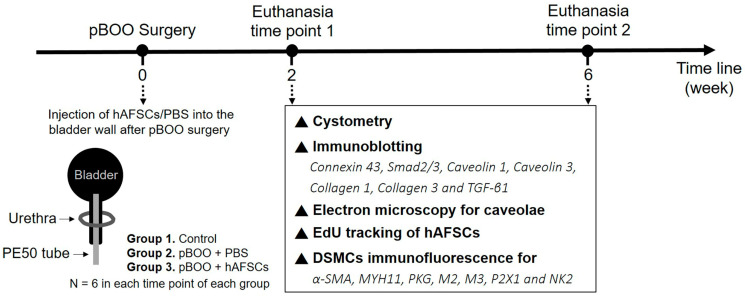
Experimental procedure. pBOO = partial bladder outlet obstruction; hAFSCs = human amniotic fluid stem cells; PBS = phosphate-buffered saline; TGF-β1 = transforming growth factor-β1.

## Data Availability

The data presented in this study are available on request to the corresponding author.

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
