# Peer review of "Local Injection of Stem Cells Can Be a Potential Strategy to Improve Bladder Dysfunction after Outlet Obstruction in Rats"

_ijms, 2024, doi:10.3390/ijms25158310_

Round 1

Reviewer 1 Report

Comments and Suggestions for Authors

Dear Author,

I give you my comment to address in your manuscript. Please add a schematic  diagram in the introduction section with an explanation of the novelty of the work.

Major Comments:

1)      Objective Clarity and Specificity: The research paper does an excellent job of outlining the background of bladder outlet obstruction (BOO) and the possible use of stem cells derived from human amniotic fluid (hAFSCs). However, the abstract does not provide a specific hypothesis or research topic. Clarity and focus might be improved by a more precise definition of the study's objectives, such as "This study aims to investigate whether hAFSCs can improve bladder function in pBOO rats by targeting specific cellular pathways."

2)      Explanation of Mechanisms: Although the study lists several molecular markers and pathways (such as connexin 43, TGF-β1, and Smad2/3), neither their involvement in bladder dysfunction nor the potential effects of hAFSCs on these pathways are briefly explained in the abstract. The abstract might be strengthened by providing a succinct description of the criteria for choosing these specific markers and how they relate to bladder dysfunction.

3)      Results and Interpretation: A summary of the findings, including altered bladder function metrics and elevated expressions of specific proteins, is given in the abstract. It would benefit from a more comprehensive synopsis of these findings' implications on the central premise. To assist readers in comprehending the relevance of the findings, for example, an explanation of the correlation between better bladder function and the observed changes in protein expression may be provided.

Minor Comments:

1)      Methodology Details: Cytometry, electron microscopy, immunofluorescence, and Western blot analysis are all mentioned in passing in the abstract. A phrase outlining the treatment groups and the overall experimental timeline—such as the length of the hAFSCs' treatment and the precise time points for analysis—would help set the research design in perspective.

2)      Statistical Significance: Although changes in several markers and bladder function metrics are reported in the abstract, it is unclear whether these changes were statistically significant. It would strengthen the findings' credibility to include assertions regarding statistical significance (such as "significant increases" or "no significant difference").

3)      Future Implications: A quick discussion of how these findings affect clinical practice could enhance the abstract. For instance, a phrase outlining the potential applications of these findings for human therapies, or the direction of future research would give the study's significance a more comprehensive perspective.

Best Regards

Reviewer 2 Report

Comments and Suggestions for Authors

Lee et al. studied about Bladder Dysfunction and stem cell injection. It is interesting.

Major

Author should highlight the novel findings in this study and should discuss about therapeutic mechanism of stem cells compared to the previous studies.

Author should include the characterization of hAFSCs.

The purpose of tracking of implanted hAFSCs in Figrue 4 should be explained and discussed.

Minor

In each result section, author should describe the purpose of the experiments.

In the first discussion section, author should summary or describe the novel findings of this paper.

Reviewer 3 Report

Comments and Suggestions for Authors

The manuscript entitled “Local Injection of Stem Cells Can Be a Potential Strategy to Improve Bladder Dysfunction after Outlet Obstruction in Rats” has described the efficiency of local injection of human amniotic fluid 24 stem cells in improvement of bladder outlet obstruction (BOO) in rats. It is a very interesting paper; however the quality of this study can be improved following addressing the following issues/questions.

·         Why did you use only female subjects in this study, considering that bladder outlet obstruction is more common in males?

·         In the Materials and Methods section, under section 4.1, what does the “(3)” in “(3) pBOO rats” refer to?

·         Where were the stem cell injections administered? Was it a single injection site? Which part of the bladder was targeted? How can you confirm that the cells were injected into the correct location?

·         What happened to the grafted hAFSCs in the bladder, given that they gradually disappeared from day 1 to day 42? What do you believe caused their disappearance?

·         In Figure 5, α-SMA is merged with DAPI, but MYH11, PKG, M2, M3, P2X1, and NK2 are not. Is there a reason for this?

·         In Figure 5, it is mentioned that a 488 secondary antibody was used, but the color appears red. Was the color changed during image analysis, or was a different type of secondary antibody used?

·         In Figure 5, while I am aware that some researchers use fluorescent optical density to quantify protein expression, I believe that immunofluorescence is not a robust method for protein quantification. I suggest using Western Blotting or other alternative techniques. However, it is still useful to have representative images from immunofluorescence staining to show differences.

·         I could not find Supplementary Table 1 and Supplementary Table 2 in the document I downloaded for review. Could you provide these tables?

Round 2

Reviewer 2 Report

Comments and Suggestions for Authors

There are no comments.

Author Response

We noticed that there is no comments from you. We thank you for your review.